# BAQ: Efficient Bit Allocation Quantization for Large Language Models

## Abstract

Post-training model quantization is a widely adopted technique for reducing the memory and computational costs of large language models (LLMs). However, most existing methods either fix a uniform bitwidth or rely on binary sensitivity groupings ("sensitive" vs. "non-sensitive") that treat all weights within a group identically, ignoring how sensitive each weight actually is and leaving the degree of sensitivity under-exploited. To address this, for the first time in the neural network quantization literature, we introduce an explicit loss–bitwidth relation that links layer-output distortion to the assigned precision, together with a sensitivity-guided bit-allocation quantization (BAQ) framework. Under mild assumptions, this modeling makes the layer-wise loss an explicit function of quantization bitwidth and yields a *convex* resource-allocation problem with a *closed-form* solution that adapts precision across weights. This choice is theoretically motivated by rate–distortion theory and validated by extensive simulations. Inspecting the solution of the proposed resource-allocation problem provides several insights (such as the equal-loss structure), which are then exploited to design the proposed algorithm. The proposed algorithm achieves a good trade-off between loss minimization and complexity and allows BAQ to be integrated into standard quantization pipelines with minimal overhead. Experimental results show that BAQ consistently outperforms GPTQ, achieving up to $56\times$ lower perplexity at the same bitwidth on large language models (e.g., OPT, Llama) ranging from 125M to 30B parameters. Leveraging our analytical results derived from solving the optimal bit allocation problem, we also provide a theoretical explanation for the observed gains.

## 1 Introduction

Large language models (LLMs) have achieved remarkable success across a wide range of natural language processing tasks (OpenAI). However, their immense scale poses significant challenges for deployment in resource-constrained environments. Model quantization is one of the key techniques to compress large neural networks (NNs) and thus for mitigating memory and compute costs. The current literature shows how it is now possible to quantize large NN parameters with low-bit representations (e.g., INT4) while maintaining good performance (Dettmers et al., 2023)(Frantar et al., 2023)(Wang et al., 2023).

While recent post-training quantization (PTQ) methods such as GPTQ (Frantar et al., 2023) leverage second-order information (e.g., proxy Hessians) to guide weight rounding, they typically operate at a fixed bitwidth, leaving *how to optimally allocate a given bit budget across weights* largely unaddressed. Weight sensitivity is considered in recent mixed-precision approaches (e.g., AWQ(Lin et al., 2024), SpQR(Dettmers et al., 2024)), but mainly in a *binary* manner by classifying weight columns as "sensitive" vs. "non-sensitive" and treating all members of a group identically, thereby *under-exploiting the degree of sensitivity* and still overlooking the optimal allocation problem.

In this paper, we propose **BAQ** (Bit Allocation Quantization), a principled bit-allocation framework for weight-only PTQ that explicitly minimizes the expected loss introduced by quantization under a global bit budget. *For the first time in neural network quantization*, we introduce an explicit per-unit loss model $L_{ij}(R_{ij})$ that links distortion to the assigned bitwidth, turning the layer-/component-wise loss into an explicit function of the bit-allocation vector. The modeling choice is theoretically motivated by rate-distortion theory, and yields a *convex* resource-allocation problem with a *closed-*

*form* solution that adapts precision across weights. Interestingly, we show that this optimal allocation satisfies an *equal-loss principle*, where each component/block contributes equally to the overall quantization loss. This property not only provides insight into the nature of optimal precision assignments, but also serves as a useful tool for designing loss-controlled quantization algorithms.

For deployment, we use a column-wise scheme (one bitwidth per weight column) that assigns bits as a monotone function of the measured sensitivity and calibrates a single global level so the total bits meet the budget. This rule is a lightweight, drop-in replacement for the fixed-bit module in GPTQ-style pipelines, with negligible metadata (e.g., 16 bits per weight column to indicate its bitwidth). Compared with prior sensitivity-based methods that use coarse (binary) groupings (e.g., "sensitive" vs. "non-sensitive") and treat all items in a group identically, BAQ fills the allocation gap by assigning bits *proportionally to the measured degree of sensitivity*, which is crucial in tight budgets (notably 2-bit). We validate BAQ on the OPT and LLaMA families at 2- and 3-bit, showing consistent perplexity improvements over existing PTQ methods.

**Contributions.** Beyond proposing yet another quantization scheme, we provide a *novel analytical framework* BAQ for bit allocation that both guides algorithm design and yields explainable results. *First*, inspired by rate-distortion theory, we introduce a per-unit loss–bitwidth mapping that makes the dependency of layer output loss on quantization precision explicit. *Second*, we formulate the bit allocation as a convex optimization problem and derive a closed-form solution that allocates bits *proportionally to the degree of sensitivity*, in contrast to heuristic, category-based (binary) bitwidth assignments. *Third*, we demonstrate consistent perplexity/accuracy improvements over GPTQ and mixed-precision baselines by extensive experiments on OPT and LLaMA families , and provide predictive explainability for these observed gains.

## 2 RELATED WORKS

**Post-Training Quantization (PTQ).** PTQ methods aim to convert pre-trained full-precision LLMs into low-precision formats without requiring retraining, making them highly practical for deployment. These techniques are generally categorized into weight-only quantization, weight-activation quantization (Xiao et al., 2023)(Shao et al., 2024), and KV (for Key and Value) cache quantization (Hooper et al., 2024). Our work focuses on the weight-only PTQ setting. In this setup, notable methods such as GPTQ (Frantar et al., 2023), QuIP (Chee et al., 2023), and AWQ (Lin et al., 2024) achieve high accuracy by leveraging Hessian-informed loss approximations or outlier-aware scaling strategies. These approaches often rely on estimating a proxy Hessian matrix from calibration data to guide quantization. GPTQ, for instance, uses an Optimal Brain Compression framework with inverse Hessian updates to minimize second-order loss. QuIP further enhances this by enforcing incoherence between weights and the Hessian. Other PTQ works like OWQ (Lee et al., 2024), SqueezeLLM (Kim et al., 2023), and SpQR (Dettmers et al., 2024) adopt sensitivity-based heuristics for identifying important weights and allocating bits accordingly. However, these approaches typically use fixed or coarse, rule-based assignments that give the same precision to all weights within a category (e.g., "salient" vs. "non-salient"). In contrast, BAQ provides a theoretically grounded mechanism to derive optimal bit assignments based on a convex formulation of Hessian-weighted quantization loss.

**OBS-based Compression.** The Optimal Brain Surgeon (OBS) framework (Hassibi et al., 1993) and its precursor OBD (LeCun et al., 1989) laid the foundation for second-order model compression by quantifying the impact of weight removal on the loss function and compensating for it through updates to remaining parameters. This foundational principle has inspired a variety of pruning and quantization techniques that leverage Hessian matrix information to guide compression decisions. Notably, GPTQ (Frantar et al., 2023) extends this paradigm to post-training quantization by minimizing a second-order Taylor expansion of the loss. SparseGPT (Frantar & Alistarh, 2023), OBC (Frantar & Alistarh, 2022), and BiLLM (Huang et al., 2024) further generalize OBS methodology to structured sparsity, joint quantization-pruning, and binary quantization, respectively. Our work draws from this second-order perspective but shifts the focus from selecting or modifying weights to allocating bitwidths under a global bit budget constraint. By minimizing a Hessian-weighted distortion objective, BAQ introduces a low-overhead bit allocation strategy that enhances the efficiency of OBS-based quantization pipelines.

## 3 PROBLEM FORMULATION

The main problem under consideration in this paper is to quantize the weights of a large NN model. For the sake of clarity and following related papers such as (Frantar et al., 2023), the layer or component index will be removed from the notation but the considered operations can be performed for any layer or any component of the NN. The focus will be primarily on feedforward architectures but the results may potentially extend to models incorporating feedback loops. Denote by $M \geq 1$ and $N \geq 1$ the respective sizes of the layer output and input. The weight matrix associated with the layer under consideration is denoted by $\boldsymbol{W} \in \mathbb{R}^{M \times N}$. Mainly for complexity reasons, it is assumed that each entry of $W$ is quantized with a scalar uniform quantizer and independently of the other entries. Each entry $w_{ij}$, $i\{1, ..., M\}$, $j\{1, ..., N\}$, of the weight matrix is thus approximated by its quantized version $\widehat{w}_{ij} = Q_{ij}(w_{ij})$, $Q_{ij}$ being a scalar uniform quantization function. The number of bits (referred to as the bitwidth) assigned to the quantizer $Q_{ij}$ is denoted by $R_{ij}$. As well motivated by previous works such as (Hassibi et al., 1993) and (Frantar et al., 2023), a relevant loss function to be considered for quantizing the weight $w_{ij}$ is as follows:

$$L_{ij} = \frac{(w_{ij} - Q_{ij}(w_{ij}))^2}{[\mathbf{H}_F^{-1}]_{n_{ij} n_{ij}}}, \tag{1}$$

where $\mathbf{H}_F$ is a proxy Hessian matrix corresponding to unquantized weights. Let $\mathbf{X} \in \mathbb{R}^{N \times P}$ be the input activation matrix to the layer, where $P$ is the number of calibration samples. Then $\mathbf{X}_F \subset \mathbf{X}$ denotes the submatrix formed by selecting the rows corresponding to unquantized weights, and the Hessian matrix corresponding to unquantized weights is approximated as $\mathbf{H}_F = 2\mathbf{X}_F \mathbf{X}_F^\top$. The diagonal entry $[\mathbf{H}_F^{-1}]_{n_{ij} n_{ij}}$ quantifies the sensitivity of the loss with respect to perturbations in $w_{ij}$, where $n_{ij}$ denotes the index of $w_{ij}$ among the unquantized weights in row $i$.

In the existing literature, the bitwidths $R_{ij}$ are typically chosen to be identical for all weights. However, empirical results on OPT models (Chee et al., 2023) indicates that quantization noise is more influential in terms of NN final performance for some weights. Therefore, there is an incentive to adapt $R_{ij}$ to the weight to be quantized. One of the motivations of this paper is precisely to formulate and solve an optimization problem which produces the optimal bitwidths to be used to quantize the NN. By optimality, it is meant in terms of the global loss associated with the considered layer that is, $L = \sum_{i=1}^{M} \sum_{j=1}^{N} L_{ij}$. One of the major difficulties in doing so is that each component $L_{ij}$ depends on $R_{ij}$ in a non-explicit and complicated way. To circumvent this difficulty, we propose the following approximation:

$$L_{ij}(R_{ij}) \approx \frac{1}{[\mathbf{H}_F^{-1}]_{n_{ij} n_{ij}}} \cdot \frac{\Delta_{ij}^2}{12} = \frac{1}{[\mathbf{H}_F^{-1}]_{n_{ij} n_{ij}}} \cdot \frac{(w_{ij}^{\max} - w_{ij}^{\min})^2}{12 \cdot 2^{2R_{ij}}} \tag{2}$$

where: $w_{ij}^{\max}$ and $w_{ij}^{\min}$ denote the maximum and minimum bounds of the quantizer $Q_{ij}$, $\Delta_{ij} = \frac{w_{ij}^{\max} - w_{ij}^{\min}}{2^{R_{ij}}}$ is the quantization step for the uniform scalar quantizer $Q_{ij}$. To build this approximation, the rationale is as follows. First, to allocate a bitwidth to a given weight, one considers the mean of the loss $L_{ij}$ instead of the loss itself. Second, we exploit a high-resolution approximation of this mean. Indeed, it is known from (Gray & Neuhoff, 1998)(Cover, 1999) that the distortion for a scalar uniform quantizer can be approximated in the high-resolution regime by $\frac{\Delta^2}{12}$, $\Delta$ being the quantization step. Remarkably, as all our simulations have shown, this approximation remains relevant even when the bitwidth is intermediate or low, which is also a behavior observed in image compression (Taubman et al., 2002).

Finally, we introduce a total bit budget for the considered layer (or component of the NN), denoted by $R_{\text{sum}}$. In practice, this constraint is key, for example, to enable the comparison of two model compression techniques using the same resources, or to impose a given total size on the model (or one of its components). The bit allocation problem to be solved thus writes as:

$$\text{(OP-A)} \quad \underset{R_{11}, ..., R_{MN}}{\text{minimize}} \quad \sum_{i=1}^{M} \sum_{j=1}^{N} c_{ij} \cdot 2^{-2R_{ij}} \tag{3}$$

$$\text{subject to} \quad \sum_{i=1}^{M} \sum_{j=1}^{N} R_{ij} \leq R_{\text{sum}}, \quad R_{ij} \geq 0, \ \forall i, j \tag{4}$$

where

$$c_{ij} = \frac{(w_{ij}^{\max} - w_{ij}^{\min})^2}{12 \cdot [\mathbf{H}_F^{-1}]_{n_{ij} n_{ij}}}.$$

In the proposed formulation, note that $R_{ij}$ is not required to be an integer. Therefore, the optimization problem (OP-A) describes a relaxed version of the initial problem. In practice, (OP-A) is solved, and the entries of the bit allocation vector are rounded according to a rule to be defined; one possible rule is provided in the next section. The purpose of the next section is to solve the relaxed problem and provide algorithms suitable for implementation.

## 4 ANALYTICAL SOLUTION AND ALGORITHM

### 4.1 OPTIMAL SOLUTION

In (OP-A), the bitwidth $R_{ij}$ for weight $w_{ij}$ is assumed to be continuous, and (OP-A) can be checked to be a convex problem which is strongly dual. The optimal solution can be proved to be as follows (Gersho & Gray, 2012):

$$R_{ij}^\star = \max\left(0, \frac{1}{2}\log_2\left(\frac{c_{ij}}{\lambda}\right)\right), \tag{5}$$

where $\lambda \propto \prod_{i=1}^{MN} c_{ij}^{1/MN}$ is a normalization factor determined by enforcing the budget constraint. By imposing $\lambda$ to meet the latter constraint and assuming that $\frac{R_{\mathrm{sum}}}{MN} \geq \max_{(i,j)} \frac{1}{2}\left[-\log_2 \frac{c_{ij}}{\left(\prod_{(i,j)} c_{ij}\right)^{1/MN}}\right]$, the interior solution can be rewritten as:

$$R_{ij}^\star = \frac{1}{2}\left(\log_2 \frac{c_{ij}}{\left(\prod_{(i,j)} c_{ij}\right)^{1/MN}}\right) + \frac{R_{\mathrm{sum}}}{MN}. \tag{6}$$

Three interesting observations can be made. First, the obtained solution is markedly different from the uniform bit allocation rule (namely, the rule used by state-of-the art solutions such as GPTQ) when the $c_{ij}$ vary widely. By inspecting the expression of $c_{ij}$, it is seen that this typically happens when the weight have different ranges or the eigenvalues of the matrix $\mathbf{H}$ are very different (which has been observed in (Chee et al., 2023)). Second, it can be checked that the optimal interior solution satisfies the equal-loss principle:

$$c_{ij} \cdot 2^{-2R_{ij}^\star} = c_{k\ell} \cdot 2^{-2R_{k\ell}^\star}, \quad \forall (i,j), (k,\ell). \tag{7}$$

Third, the total quantization loss under the optimal allocation $R_{ij}^\star$ is:

$$\sum_{i,j} c_{ij} \cdot 2^{-2R_{ij}^\star} = MN \left(\prod_{i,j} c_{ij}\right)^{\frac{1}{MN}} \cdot 2^{-2R_{\mathrm{sum}}/MN}, \tag{8}$$

while uniform allocation yields a loss proportional to the arithmetic mean of $\{c_{ij}\}$. Thus, the relative gain of optimal over uniform allocation is:

$$\frac{\mathrm{Loss}_{\mathrm{optimal}}}{\mathrm{Loss}_{\mathrm{uniform}}} = \frac{\left(\prod_{i,j} c_{ij}\right)^{1/MN}}{\frac{1}{MN}\sum_{i,j} c_{ij}}. \tag{9}$$

This shows that larger variance in $\{c_{ij}\}$ leads to greater improvement from optimal bit allocation, aligning with the equal-loss principle and justifying BAQ's efficiency.

This structure is exploited for the design of the following practical algorithm. More precisely, the practical weight quantization technique we propose consists of three sub-algorithms. The motivation behind these algorithms is twofold: to trade off between performance gains and complexity; to allow the proposed allocation rule to be integrated in existing model compression techniques. The following three subsections describe the three proposed algorithms which allow the proposed quantization

technique to be implemented for large NN models such as LLMs. Leveraging the results established so far we will introduce the **BAQ** algorithm, which efficiently assigns bitwidths to each column of the weight matrix $W$ based on Hessian-informed sensitivity. The algorithm is grounded in the *equal-loss principle*, which suggests that optimal quantization is achieved when each column of $\boldsymbol{W}$ contributes equally to the total loss.

## 4.2 COLUMN-WISE BIT ALLOCATION

Instead of assigning a distinct number of quantization bits to each individual weight, we consider a simplified bit allocation scheme, in which a shared bitwidth is assigned to each entry of the column of $\boldsymbol{W}$. This approach is motivated by two key observations. First, the approximation of quantization error by its expected value becomes more accurate at the column level, since the total quantization error aggregates over multiple weights. Second, the bitwidth overhead required to encode per-weight precision can be substantially reduced by sharing bitwidths across larger structures. The overhead aspect is key for quantizing large models. Therefore, we introduce a reference quantization loss value $L_{\text{ref}}$, such that each column of $\boldsymbol{W}$ is quantized to ensure its individual loss equals $L_{\text{ref}}$. Given the column sensitivity coefficient $C_j$ defined as:

$$C_j = \sum_{i=1}^{M} c_{ij} = \sum_{i=1}^{M} \frac{(w_i^{\max} - w_i^{\min})^2}{12 \cdot [\mathbf{H}_F^{-1}]_{q_{ij}q_{ij}}}, \tag{10}$$

the required bitwidth for every entry of column $j$, denoted by $R_j$ and imposed to satisfy the equal-loss equality $C_j \cdot 2^{-2R_j} = L_{\text{ref}}$, is given by:

$$R_j = \frac{1}{2} \log_2 \left( \frac{C_j}{L_{\text{ref}}} \right). \tag{11}$$

In practice, we round $R_j$ to the nearest integer to obtain a hardware-friendly bit assignment and also ensure $R_j$ is non-negative. The following procedure summarizes this bit allocation strategy:

---

**Algorithm 1** Bit Allocation Given Reference Loss

---

**Require:** Sensitivity coefficients $\{C_j\}_{j=1}^{N}$, reference loss $L_{\text{ref}}$
**Ensure:** Integer bit allocations $\{R_j\}_{j=1}^{N}$
 1: **for** each column $j = 1$ to $N$ **do**
 2:     $R_j \leftarrow \frac{1}{2} \log_2 \left( \frac{C_j}{L_{\text{ref}}} \right)$
 3:     $R_j \leftarrow \max(0, \text{round}(R_j))$
 4: **return** $\{R_j\}_{j=1}^{N}$

---

## 4.3 LAYER-WISE REFERENCE LOSS ESTIMATION

As shown in equation (8), the total quantization loss for a layer is exponentially dependent on the average bitwidth. This observation enables us to estimate an appropriate $L_{\text{ref}}$ for each layer based on a desired average bitwidth $R_{\text{ref}}$. Specifically, we begin by selecting an initial value $L_{\text{init}}$, which corresponds to an initial average bitwidth $R_{\text{init}}$ computed via Algorithm 1. To align the final bit allocation with a target average bitwidth $R_{\text{ref}}$, we adjust the reference loss according to:

$$L_{\text{ref}} = L_{\text{init}} \cdot 2^{2(R_{\text{init}} - R_{\text{ref}})}, \tag{12}$$

where $L_{\text{init}}$ is an initial reference loss corresponding to an empirically estimated average bitwidth $R_{\text{init}}$. This formulation ensures that the resulting bit allocation is centered around the desired average $R_{\text{ref}}$.

It is important to note that different layers often exhibit vastly different sensitivity distributions and quantization characteristics. As a result, achieving the same average bitwidth across layers typically requires setting distinct values of $L_{\text{ref}}$ for each layer. Empirical observations confirm that using a fixed global $L_{\text{ref}}$ can lead to substantial discrepancies in layer-wise average bitwidths $R_{\text{avg}}$, which in

turn causes notable degradation in model performance. To ensure consistent accuracy and reliable compression, it is essential to control $R_{\text{avg}}$ within a narrow range across layers.

---

**Algorithm 2** Reference Loss Estimation for Target Average Bitwidth

---

**Require:** Sensitivity coefficients $\{C_j\}_{j=1}^N$, initial reference loss $L_{\text{init}}$, target average bitwidth $R_{\text{ref}}$
**Ensure:** Updated reference loss $L_{\text{ref}}$
 1: Use Algorithm 1 with $L_{\text{init}}$ to compute $\{R_j\}_{j=1}^N$
 2: $R_{\text{init}} \leftarrow \frac{1}{N} \sum_{j=1}^N R_j$
 3: $L_{\text{ref}} \leftarrow L_{\text{init}} \cdot 2^{2(R_{\text{init}} - R_{\text{ref}})}$
 4: **return** $L_{\text{ref}}$

---

### 4.4 FULL BAQ WORKFLOW AND INTEGRATION WITH EXISTING QUANTIZATION TECHNIQUES

The BAQ algorithm described above can be readily integrated into existing quantization pipelines by replacing their static or heuristic bit assignment with our sensitivity-guided bit allocation strategy. In particular, methods such as GPTQ (Frantar et al., 2023), which apply fixed-bit quantization, can benefit significantly from our layer-wise adaptive bitwidth assignment.

To demonstrate this, we present the full BAQ workflow as a drop-in replacement for the bit allocation module in methods based on GPTQ.

---

**Algorithm 3** Full BAQ Workflow with GPTQ-based Quantization

---

**Require:** Weight matrix $W \in \mathbb{R}^{M \times N}$, inverse Hessian diagonal $\{[\mathbf{H}_F^{-1}]_{q_{ij}q_{ij}}\}$, target average bitwidth $R_{\text{ref}}$, initial reference loss $L_{\text{init}}$
**Ensure:** Quantized weights $\widehat{W}$
 1: Compute sensitivity coefficients: $C_j = \sum_{i=1}^M \frac{(w_i^{\max} - w_i^{\min})^2}{12 \cdot [\mathbf{H}_F^{-1}]_{q_{ij}q_{ij}}}$
 2: Estimate refined reference loss $L_{\text{ref}}$ using Algorithm 2
 3: Compute optimal bitwidths $\{R_j\}$ using Algorithm 1 with $L_{\text{ref}}$
 4: **for** each column $j = 1$ to $N$ **do**
 5:     Quantize column $W_{:,j}$ using GPTQ with bitwidth $R_j$
 6: **return** Quantized weight matrix $\widehat{W}$

---

For other model compression methods, as long as the loss induced by weight compression can be explicitly expressed, our BAQ framework can be similarly applied. The key difference lies in how the sensitivity coefficients $C_j$ are calculated, which may vary depending on the specific compression strategy employed. For instance, pruning-based or low-rank approximation methods may define $C_j$ using different second-order metrics or task-specific criteria. Nonetheless, once a meaningful per-component loss approximation is available, BAQ provides a general mechanism to optimally allocate quantization precision under a global budget.

## 5 EXPERIMENTS

**Overview.** We evaluate BAQ on the OPT (Zhang et al., 2022) and Llama2/3 model families across a wide range of sizes (from 125M to 30B parameters), focusing exclusively on aggressive 2-bit/3-bit weight-only quantization. Classical methods such as GPTQ are known to perform well in the 4-bit setting, but often degrade significantly at very low bits. In contrast, BAQ enables high-accuracy quantization even at low precision across diverse models. We show that (1) BAQ consistently outperforms existing methods on both perplexity and accuracy metrics across diverse datasets, especially at 2-bit regime, and (2) these gains primarily stem from BAQ's ability to allocate bits more efficiently by exploiting the heterogeneous sensitivity of individual weights.

**Setup.** Our experimental setup follows the GPTQ pipeline (Frantar et al., 2023), and we use HuggingFace implementations of all models. We quantized all models using a single NVIDIA A100

GPU with 80GB of memory. Our calibration dataset consists of 128 randomly sampled 2048-token segments from the C4 dataset (Raffel et al., 2020), without any retraining or task-specific tuning. We report perplexity on WikiText2 (Merity et al., 2016), PTB (Marcus et al., 1994), and C4, and zero-shot accuracy on StoryCloze (Mostafazadeh et al., 2016), PIQA (Tata & Patel, 2003) and ARC-Easy (Boratko et al., 2018). We use structured allocation to ensure each weight column uses the same number of bits.

**Methods.** We compare BAQ with GPTQ/AWQ/SPQR across OPT and Llama2/3 models. In all cases, BAQ uses our closed-form allocation rule (Section 4) for bit assignment.

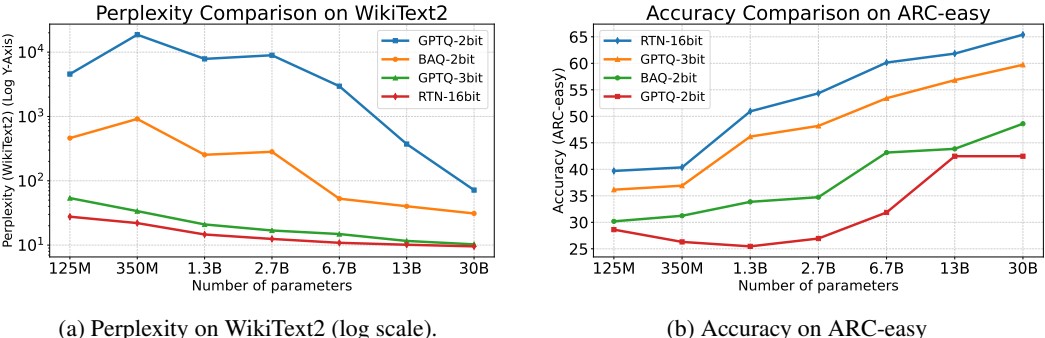

(a) Perplexity on WikiText2 (log scale).

(b) Accuracy on ARC-easy

Figure 1: Comparison between BAQ-2bit and several baselines (GPTQ-2bit, GPTQ-3bit, RTN-16bit) on two representative tasks: WikiText2 (left) and ARC-easy (right). BAQ-2bit consistently improves over GPTQ-2bit.

Table 1: Comparison of BAQ and GPTQ on various OPT models and datasets with **2/3-bit quantization**. Perplexity (↓) and accuracy (↑) metrics are reported.

| Method | Model | Avg. Bits | Perplexity (↓) | | | Accuracy (↑) | | |
|---|---|---|---|---|---|---|---|---|
| | | | C4 | WikiText2 | PTB | SC | PIQA | ARC-E |
| BAQ | Llama2-7B | 1.99 | 430.78 | 873.39 | - | 50.03 | 52.67 | 27.89 |
| GPTQ | Llama2-7B | 2.00 | 2.2e3 | 1.1e4 | - | 50.03 | 52.18 | 25.26 |
| AWQ | Llama2-7B | 2 | 2.2e5 | 1.7e5 | - | 49.98 | 52.39 | 24.75 |
| SPQR | Llama2-7B | 2.1 | 1.2e3 | 3.7e3 | 2.2e4 | 49.01 | 51.74 | 28.60 |
| BAQ | Llama2-13B | 2.01 | 71.45 | 142.86 | 1.2e3 | 53.66 | 53.75 | 28.95 |
| GPTQ | Llama2-13B | 2.00 | 293.79 | 1.0e3 | 4.4e3 | 49.97 | 51.14 | 28.25 |
| AWQ | Llama2-13B | 2 | 1.2e5 | 9.5e4 | - | - | 53.26 | 23.04 |
| SPQR | Llama2-13B | 2.1 | 292.97 | 1.0e3 | 3.4e3 | - | - | - |
| BAQ | Llama3-8B | 1.99 | 3.7e3 | 3.2e4 | 2.1e4 | - | - | - |
| GPTQ | Llama3-8B | 2.00 | 2.7e5 | 1.0e6 | 1.6e6 | - | - | - |
| BAQ | OPT-2.7B | 1.92 | 126.50 | 282.47 | 326.53 | - | 55.17 | 32.11 |
| GPTQ | OPT-2.7B | 2.00 | 4388 | 8949 | 8281 | - | 48.42 | 26.94 |
| BAQ | OPT-6.7B | 2.07 | 33.64 | **52.71** | 70.84 | - | 63.87 | 43.16 |
| GPTQ | OPT-6.7B | 2.00 | 500.7 | **2958** | 2521 | - | 55.11 | 31.86 |
| BAQ | OPT-30B | 1.95 | 24.21 | 31.05 | 47.98 | - | 66.97 | 48.6 |
| GPTQ | OPT-30B | 2.00 | 29.59 | 71.7 | 88.19 | - | 66.05 | 42.47 |
| BAQ | Llama2-7B | 2.98 | 10.90 | 9.26 | 107.86 | 72.37 | 71.98 | 59.30 |
| GPTQ | Llama2-7B | 3.00 | 10.39 | 9.50 | 7.3e3 | 71.51 | 70.78 | 60.18 |
| AWQ | Llama2-7B | 3.00 | 23.85 | 24.00 | - | - | 65.02 | 52.78 |
| SPQR | Llama2-7B | 3.10 | 9.35 | 7.68 | 48.91 | 73.86 | 74.37 | 67.37 |
| BAQ | OPT-6.7B | 3.00 | 16.78 | 16.25 | 22.32 | 70.92 | 73.07 | 60.88 |
| GPTQ | OPT-6.7B | 3.00 | 15.41 | 15.14 | 18.46 | 69.86 | 73.01 | 59.47 |

**Main results**. As illustrated in Figure 1, with 2-bit quantization, BAQ consistently largely outperforms existing methods (such as GPTQ, AWQ) across model scales on both perplexity and accuracy, demonstrating superior performance in language modeling and zero-shot reasoning tasks. Table 1 presents a more detailed comparison between the proposed BAQ method and GPTQ across a range of OPT models and evaluation tasks. Two metrics are reported: perplexity and accuracy. Perplexity

measures how well the model predicts the next token in a sequence—lower values indicate better language modeling performance. Accuracy, in this context, refers to performance on zero-shot multiple-choice benchmarks such as PIQA, and ARC-easy. These tasks assess the model's reasoning and language understanding capabilities without any task-specific fine-tuning, and accuracy is computed as the fraction of correct choices made over the evaluation set.

Several key observations emerge. First, BAQ consistently outperforms GPTQ in perplexity across all OPT model sizes, indicating better preservation of token-level predictions under aggressive 2-bit quantization. Second, BAQ yields substantial accuracy gains on downstream zero-shot tasks, particularly for larger models for 2-bit regime. Third, for 3-bit quantization, BAQ have comparable results to state-of-the-art methods. In this higher-bit regime, the overall resource budget is already less constrained, so the relative gain from optimal allocation is smaller. Moreover, BAQ's theoretical advantage is partially offset by practical constraints, such as rounding continuous allocations to integer bitwidths and using standard quantization approximations. Overall, these results validate the core design of BAQ: bitwidths are allocated according to Hessian-based sensitivity to preserve model semantics under quantization. The strong gains in both perplexity and accuracy, especially in low-bit regimes, highlight BAQ's effectiveness as a scalable, high-performance solution for post-training quantization.

**Running time**. BAQ incurs approximately 1.5× running time compared to GPTQ, mainly due to the additional computation for layer-wise bit allocation.

Table 2: Running time (in seconds) of GPTQ and BAQ for different OPT models.

| Method | OPT-2.7B | OPT-6.7B | OPT-13B | OPT-30B |
|---|---|---|---|---|
| GPTQ | 517.08 | 1194.45 | 2423.57 | 6220.92 |
| BAQ | 797.07 | 1734.22 | 3336.18 | 8211.75 |

**Bitwidth histogram**. To further understand the effectiveness of BAQ-2bit, we analyze the bit allocation profile on OPT-2.7B, visualized in Figure 2. The figure plots the distribution of allocated bitwidths across all weights in the model. While the majority of weights are still assigned the minimal 2-bit representation, a non-negligible portion of weights are allocated to lower or higher precision. This illustrates that BAQ successfully integrates the bit allocation mechanism into the quantization process, assigning fewer bits to more robust weights and more bits to sensitive ones.

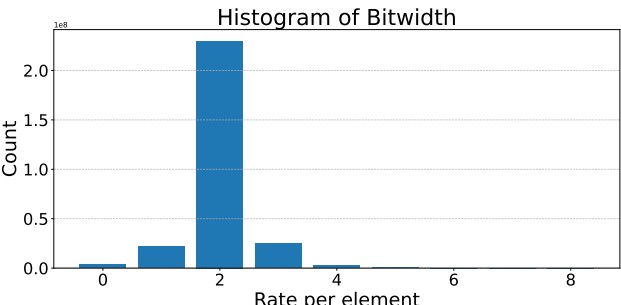

Figure 2: Distribution of assigned bitwidths $R_j$ in OPT-2.7B under BAQ. While most weights are quantized to 2 bits, BAQ adaptively allocates higher or lower precision to match weight sensitivity. Layers with higher variance in sensitivity (as measured by $\{C_j\}$) exhibit broader bitwidth distributions, reflecting structural adaptivity.

**Layer-wise loss analysis**. Fig. 3 compares two metrics across transformer layers to explain the effectiveness of BAQ from the layer-wise loss perspective: *Ratio_L*, the ratio of quantization loss under BAQ to that under GPTQ (uniform 2-bit), and *Ratio_C*, the geometric-to-arithmetic mean ratio of the sensitivity coefficients $\{C_j\}$. Quantization loss is approximated by $\sum_j C_j 2^{-2R_j}$, where uniform bitwidth yields loss scaling with the arithmetic mean, while optimal allocation achieves scaling with the geometric mean.

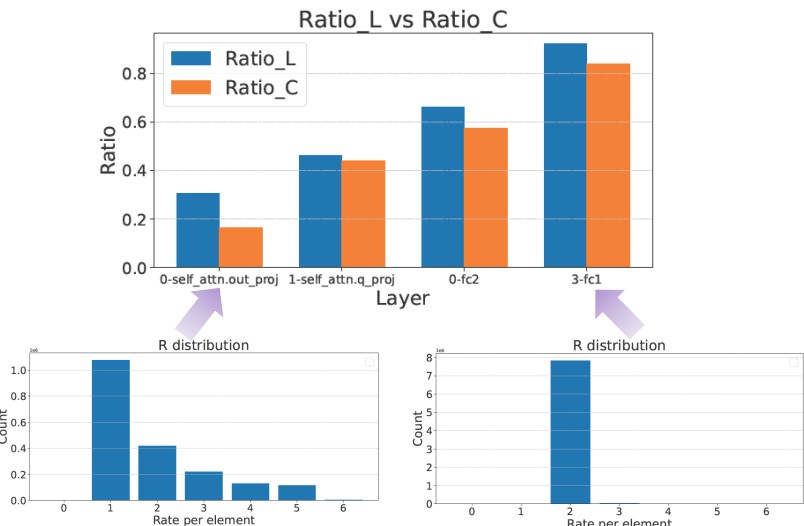

Figure 3: Layer-wise comparison of quantization efficiency with OPT-2.7B model. **Ratio_C** (geometric mean over arithmetic mean of $\{C_j\}$) characterizes the potential gain from optimal bit allocation. **Ratio_L** measures the realized gain by BAQ compared to GPTQ. Layers with more dispersed sensitivity (lower Ratio_C) benefit more from bit allocation.

Fig. 3 reveals that layers with low *Ratio_C*, which indicate diverse sensitivities on weights, show greater improvement under BAQ (lower *Ratio_L*). This confirms that bit allocation is especially effective in layers with heterogeneous sensitivity. Since $C_j$ can be computed from the diagonal entries of the Cholesky decomposition of the inverse Hessian $H^{-1}$ (Frantar et al., 2023), a large variation in $\{C_j\}$ typically indicates that the original Hessian $H$ has a wide spread of eigenvalues, meaning some directions in the weight space are much more sensitive than others. In such layers, uniform quantization inefficiently allocates bits to insensitive weights, while BAQ adaptively assigns higher precision where it matters most. This adaptivity is reflected in the greater variance of bitwidths observed in those layers.

**Additional results**. More results to illustrate the equal loss principle validation in BAQ, the integration and discussion to the transformation-based method, and the detailed additional computation overheads are shown in Appendix.

## 6 CONCLUSION

This paper introduced BAQ, a principled bit allocation framework for post-training model quantization. By formulating bit allocation as a convex optimization problem over a sensitivity-aware loss model, we derived a closed-form rule that assigns quantization precision to individual weights based on their Hessian-informed importance. The resulting algorithm is simple, efficient, and compatible with existing quantization pipelines such as GPTQ and QuIP. Experimental results demonstrate that BAQ delivers substantial gains over uniform-bit quantization. Importantly, these improvements come with negligible computational overhead, making BAQ highly practical for real-world deployment. Our analysis also revealed a strong empirical correlation between the variance in Hessian-derived sensitivity coefficients and the benefit of bit allocation, validating the theoretical underpinnings of our approach. In addition, we showed that while transformation-based methods like QuIP tend to uniformize weight sensitivities (thus limiting the gains from adaptive bitwidth), BAQ remains effective and robust even without such preprocessing. Looking forward, an interesting direction is to jointly optimize weight transformations and bit allocation to maximize their complementary strengths. Overall, BAQ offers a lightweight, theoretically grounded, and high-performance solution for aggressive quantization in modern LLMs.

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

APPENDIX

The appendices provide additional technical content that supports and extends the main paper. First, we present rigorous derivations for the optimal bit allocation problem introduced in Section 4.1, including the closed-form solution, the equal-loss principle, and the geometric-vs-arithmetic mean quantization loss ratio. Second, we demonstrate the generalizability of BAQ by applying it to more models, confirming its efficacy across different architectures. Third, we show how BAQ can be integrated into advanced transformation-based quantization methods like QuIP. Empirical results reveal that while QuIP alone benefits from incoherence-promoting transformations, combining it with BAQ yields further improvements, especially when sensitivity varies significantly across columns, highlighting the flexibility and broad applicability of BAQ across a wide range of quantization settings. At last, we provide the complexity and overhead analysis. All codes of this paper are available at https://github.com/CSU-ModelCompression/BAQ.

## A    PROOF OF OPTIMAL BIT ALLOCATION RESULTS IN SEC 4.1

In this section, we rigorously prove the key theoretical results from Section 4.1 of the main paper, including:

- the closed-form solution for the optimal bit allocation $R_{ij}^*$,
- the equal-loss property,
- and the geometric-vs-arithmetic mean ratio for quantization loss.

We consider the relaxed optimization problem:

$$\min_{\{R_{ij} \geq 0\}} \quad \sum_{i,j} c_{ij} \cdot 2^{-2R_{ij}}$$
$$\text{subject to} \quad \sum_{i,j} R_{ij} \leq R_{\text{sum}}, \tag{13}$$

### A.1    CLOSED-FORM EXPRESSION FOR $R_{ij}^*$

We define the Lagrangian:

$$\mathcal{L}(\{R_{ij}\}, \lambda) = \sum_{i,j} c_{ij} \cdot 2^{-2R_{ij}} + \lambda \left( \sum_{i,j} R_{ij} - R_{\text{sum}} \right). \tag{14}$$

Set the derivative with respect to $R_{ij}$ to zero:

$$\frac{\partial \mathcal{L}}{\partial R_{ij}} = -2 \ln(2) \cdot c_{ij} \cdot 2^{-2R_{ij}} + \lambda = 0. \tag{15}$$

Solving the above equation yields:

$$R_{ij}^* = \max \left( 0, \frac{1}{2} \log_2 \left( \frac{c_{ij}}{\lambda'} \right) \right), \quad \lambda' = \frac{\lambda}{2 \ln 2}. \tag{16}$$

If the total budget $R_{\text{sum}}$ is sufficiently large such that the optimal solution satisfies $R_{ij}^* > 0$ for all $(i, j)$, we can omit the max operator. In this case, to satisfy the constraint $\sum_{i,j} R_{ij}^* = R_{\text{sum}}$, we obtain:

$$\lambda' = \left( \prod_{i,j} c_{ij} \right)^{1/MN} \cdot 2^{-2R_{\text{sum}}/MN}. \tag{17}$$

Substituting into the expression gives:

$$R_{ij}^* = \frac{1}{2} \log_2 \left( \frac{c_{ij}}{G} \right) + \frac{R_{\text{sum}}}{MN}, \tag{18}$$

where $G = \left( \prod_{i,j} c_{ij} \right)^{1/MN}$ represents the geometry mean of $\{c_{ij}\}$.

## A.2 EQUAL-LOSS PROPERTY

From optimality:

$$2^{-2R_{ij}^*} = \frac{\lambda'}{c_{ij}} \quad \Rightarrow \quad c_{ij} \cdot 2^{-2R_{ij}^*} = \lambda'. \tag{19}$$

This result implies that all quantization-induced loss terms are equal across weights:

$$c_{ij} \cdot 2^{-2R_{ij}^*} = c_{k\ell} \cdot 2^{-2R_{k\ell}^*}, \quad \forall (i,j), (k,\ell). \tag{20}$$

## A.3 QUANTIZATION LOSS RATIO: GEOMETRIC VS. ARITHMETIC MEAN

Under optimal allocation:

$$\text{Loss}_{\text{optimal}} = \sum_{i,j} c_{ij} \cdot 2^{-2R_{ij}^*} = MN \cdot G \cdot 2^{-2R_{\text{sum}}/MN}. \tag{21}$$

Under uniform allocation $R_{ij}^{\text{uni}} = R_{\text{sum}}/MN$:

$$\text{Loss}_{\text{uniform}} = \sum_{i,j} c_{ij} \cdot 2^{-2R_{\text{sum}}/MN} = MN \cdot A \cdot 2^{-2R_{\text{sum}}/MN}, \tag{22}$$

where $A = \frac{1}{MN} \sum_{i,j} c_{ij}$ represents the arithmetic mean of $\{c_{ij}\}$. Therefore, the ratio becomes:

$$\frac{\text{Loss}_{\text{optimal}}}{\text{Loss}_{\text{uniform}}} = \frac{G}{A} = \frac{\left(\prod c_{ij}\right)^{1/MN}}{\frac{1}{MN} \sum c_{ij}}. \tag{23}$$

This confirms that the relative benefit of optimal allocation over uniform allocation increases with greater variance in $\{c_{ij}\}$.

## B EMPIRICAL VALIDATION OF THE EQUAL-LOSS PRINCIPLE

In this part, we provide an empirical check of the equal-loss principle on a representative layer.

For the second layer of the second Transformer block in LLaMA-2 7B, we compute the realized per-column loss

$$\mathcal{L}_j \triangleq C_j \, 2^{-2R_j},$$

under two precision-allocation schemes: (i) *Uniform* (GPTQ-style equal bitwidth per column) and (ii) *BAQ* (bit-allocation by our rule). We summarize the empirical distribution of $\{\mathcal{L}_j\}$ via its sample mean and variance.

Table 3: Per-column loss statistics $\mathcal{L}_j = C_j 2^{-2R_j}$ on LLaMA-2 7B (Transformer block 2, layer 2). BAQ achieves both lower mean loss and markedly smaller variance than uniform allocation at the same bit budget.

| Allocation | Mean | Variance |
|---|---|---|
| Uniform (GPTQ) | 14.89 | 3752.06 |
| BAQ | 3.98 | 1.2829 |

Two patterns are evident (Table 3): (i) the *mean* loss is substantially lower under BAQ for the same bit budget, indicating more effective precision allocation; and (ii) the *variance* of per-column losses is drastically reduced, showing that BAQ *equalizes* losses across columns—consistent with the equal-loss principle.

## C    INTEGRATION WITH TRANSFORMATION-BASED QUANTIZATION

As demonstrated in prior sections, our bit allocation algorithm provides significant improvements over fixed-bit quantization schemes by adapting bitwidths to the sensitivity of individual weight groups. A natural question arises: can BAQ further enhance advanced quantization pipelines such as QuIP, which already use transformations to improve quantization robustness? To investigate this, we integrated BAQ into QuIP by replacing its uniform bitwidth assignment with BAQ's optimal per-column bitwidths.

To study the interaction between BAQ and transformation-based quantization methods such as QuIP, we consider the use of orthogonal linear transformations applied to the weight matrix $\mathbf{W}$ and its corresponding Hessian approximation $\mathbf{H}$, as originally proposed in QuIP. Specifically, QuIP leverages transformation pairs $(\mathbf{U}, \mathbf{V})$ to map weights and curvature into an incoherent domain:

$$\mathbf{W} \mapsto \mathbf{U}^{\top}\mathbf{W}\mathbf{V}, \quad \mathbf{H} \mapsto \mathbf{V}^{\top}\mathbf{H}\mathbf{V},$$

where $\mathbf{U}, \mathbf{V}$ are blockwise orthogonal matrices.

The choice of transformation matrices $\mathbf{U}$ and $\mathbf{V}$ plays a crucial role in transformation-based quantization, as it directly affects the distribution of sensitivity coefficients $\{C_j\}$ in the transformed domain. Following the QuIP framework, we construct $\mathbf{U}$ and $\mathbf{V}$ as block-diagonal matrices composed of smaller orthogonal blocks. Specifically, we build $\mathbf{U} = \mathrm{diag}(\mathbf{U}_1, \ldots, \mathbf{U}_{N_p})$ and similarly for $\mathbf{V}$, where each $\mathbf{U}_i \in \mathbb{R}^{p \times p}$ is an orthogonal matrix.

To evaluate how the structure of these orthogonal blocks impacts quantization performance, we consider three construction strategies:

- **Mild transformation** ($\sigma = 10^{-2}$): Each block $\mathbf{U}_i \in \mathbb{R}^{p \times p}$ is constructed as the orthogonal factor $\mathbf{Q}$ from the QR decomposition of a random matrix of the form $\mathbf{I} + \sigma \cdot \mathbf{G}$, where $\mathbf{G} \in \mathbb{R}^{p \times p}$ is a matrix with i.i.d. entries sampled from $\mathcal{N}(0, 1)$. The resulting blocks are close to identity and introduce limited incoherence.

- **Moderate transformation** ($\sigma = 10^{-1}$): We apply the same procedure but increase the noise level to $\sigma = 10^{-1}$, generating blocks that are more randomized and less correlated with the identity, thereby inducing stronger incoherence.

- **Highly randomized transformation** ($\sigma \to \infty$): Each block $\mathbf{U}_i$ is drawn as a fully random orthogonal matrix of size $p \times p$, typically sampled from the Haar distribution via QR decomposition of a standard Gaussian matrix. This represents the limiting case of the above construction with very large $\sigma$. This construction achieves high incoherence between transformed features.

To assess the compatibility of BAQ with transformation-based quantization frameworks such as QuIP, we apply linear transformations $\mathbf{U}$ and $\mathbf{V}$ to the input covariance matrix $\mathbf{H}$ and the weight matrix $\mathbf{W}$, respectively, following the design of QuIP. The bitwidths in QuIP are uniformly set per column, and the quantization loss of each column is empirically measured. Using this, we estimate the sensitivity coefficients $C_j$ by rearranging the proxy loss expression $C_j 2^{-2R}$, since all columns use the same bitwidth $R$ in the QuIP baseline. This enables us to approximate the induced loss from quantizing column $j$, which forms the basis for evaluating the potential benefit of BAQ's bit allocation.

We analyze how the effectiveness of BAQ varies under different transformation settings by comparing QuIP and QuIP+BAQ across three increasingly incoherent configurations: mild, moderate, and highly randomized. As shown in Table 4, applying BAQ yields substantial perplexity reductions in the mild and moderate cases. For instance, under mild transformation, BAQ reduces the perplexity on WikiText2 from 3032.47 to 647.92. This significant gain stems from the high variance in weight sensitivities across columns, which allows BAQ to exploit bitwidth adaptation effectively.

To further interpret this behavior, we refer to the histograms of *Ratio_C* in Figure 4. *Ratio_C* represents the geometric mean to arithmetic mean ratio of sensitivity coefficients $\{C_j\}$, which approximates the percentage of loss reduction achievable by optimal bit allocation compared to uniform allocation with the same average bitwidth. As shown in the histograms, under highly randomized transformations, most *Ratio_C* values are close to 1, implying limited benefit from bit reallocation. In contrast, with moderate or mild transformations, the *Ratio_C* values exhibit greater

Table 4: **Perplexity** comparison of QuIP and QuIP+BAQ under three transformation settings (OPT-125m), with an average bitwidth of 2 bits. BAQ is applied to adjust bitwidths per-column, while preserving the same overall budget.

| Transformation | Dataset | QuIP | QuIP+BAQ |
|---|---|---|---|
| Mild | C4 | 1760.70 | 253.49 |
| | WikiText2 | 3032.47 | 647.92 |
| | PTB | 3067.05 | 590.97 |
| Moderate | C4 | 348.15 | 326.76 |
| | WikiText2 | 638.39 | 530.00 |
| | PTB | 1686.45 | 644.29 |
| Highly Randomized | C4 | 48.79 | 47.91 |
| | WikiText2 | 81.09 | 79.14 |
| | PTB | 214.09 | 223.14 |

variation and are frequently much less than 1, indicating that BAQ can yield substantial improvements. This structural variance enables BAQ to provide meaningful improvements by assigning more bits to sensitive directions.

Thus, while BAQ consistently preserves the bit budget, its relative impact depends strongly on the underlying sensitivity structure shaped by the transformation. These findings not only validate the use of *Ratio_C* as a diagnostic metric for allocation benefit, but also highlight the complementary nature of BAQ and incoherence processing.

These findings suggest that BAQ is especially advantageous in settings where incoherence is difficult to achieve or the transformation quality is uncertain. Unlike QuIP, which relies on the accurate construction and inversion of transformation matrices, BAQ provides a lightweight and robust precision scheduling mechanism that adapts to structural sensitivity without introducing additional inference-time overhead.

## D   COMPLEXITY AND OVERHEAD ANALYSIS

The BAQ algorithm brings minimal additional computational complexity, as it reuses quantities like the inverse Hessian diagonal and weight statistics already computed in standard pipelines such as GPTQ. Its core step, computing each bitwidth via equation (11), followed by rounding, involves only cheap, element-wise operations, negligible compared to matrix or calibration computations.

In terms of encoding overhead, BAQ adopts a structured quantization scheme where all weights in a column share the same bitwidth $R_j$. As such, it suffices to transmit one additional value per column to indicate the bitwidth used. Since the number of bits typically ranges from 0 to 15, a 4-bit header per column is sufficient to represent all bitwidths without loss. Given that a typical column contains approximately 1000 weight elements, the overhead is only 0.004 bits per component, which is negligible compared to the savings achieved through mixed-precision quantization.

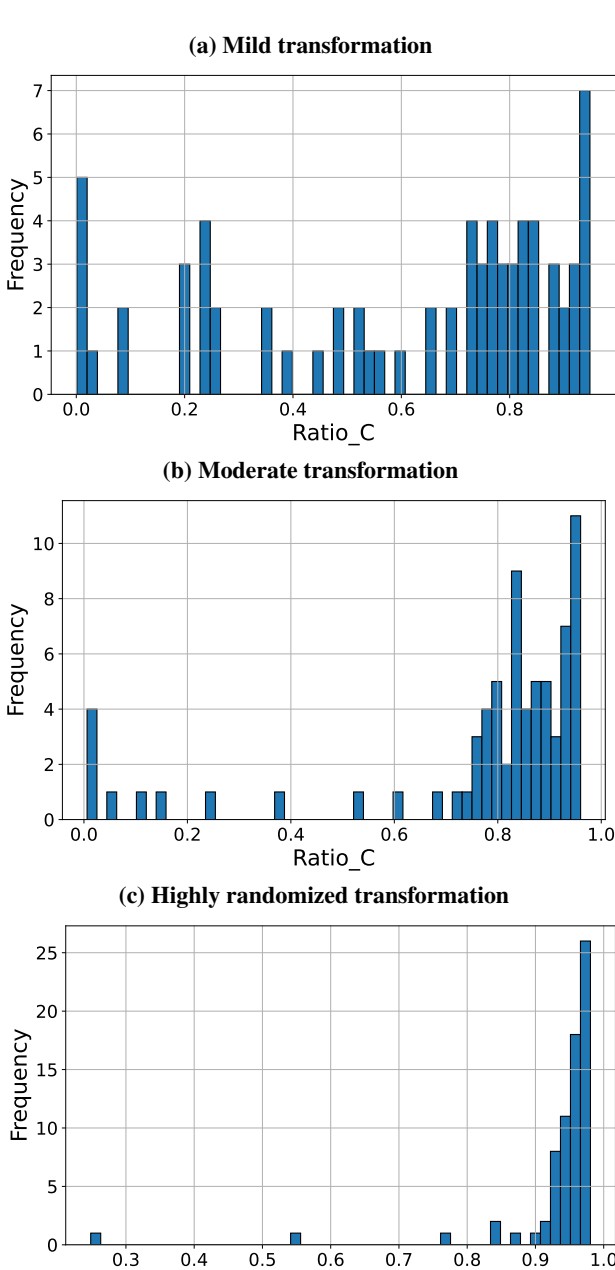

Figure 4: Histograms of $\mathrm{Ratio\_C} = \mathrm{GM}(\{C_j\})/\mathrm{AM}(\{C_j\})$ across different transformation strategies. A lower value of $\mathrm{Ratio\_C}$ indicates greater dispersion in the sensitivity coefficients $\{C_j\}$, which corresponds to higher potential gains from bit allocation. In mild and moderate transformations, $\mathrm{Ratio\_C}$ often falls significantly below 1, suggesting that BAQ can yield substantial improvement over uniform quantization. In contrast, highly randomized transformations yield $\{C_j\}$ distributions that are nearly uniform, with $\mathrm{Ratio\_C} \approx 1$, thus diminishing the relative benefit of bit allocation. These patterns align with the performance differences observed in Table 4.

