# OpenReview forum: "BAQ: Efficient Bit Allocation Quantization for Large Language Models"
_ICLR.cc/2026/Conference — ICLR 2026 Conference Withdrawn Submission_

### Official Review · Reviewer_ZNPY · 2025-10-23

**Soundness:** 2
**Presentation:** 2
**Contribution:** 2
**Rating:** 2
**Confidence:** 5

**Summary:**

In this work, the authors presented a new framework to allocate different bit-widths to each weight.

By assuming the uniform weight distribution, the authors formulated the bit allocation problem as convex optimization problem and presented its closed-form solution.

By integrating the proposed method into GPTQ, the authors showed that the performance of GPTQ can be enhanced by combining the proposed method.

**Strengths:**

The paper is well-written and easy to follow.

The bit allocation formulation seems to be correct, and the corresponding solution (mathematical derivation) is correct.

**Weaknesses:**

1. The major concern is that the reported performance is not good. From the results in Table 1, I could observe the improvement when GPTQ is combined with BAQ. However, the final perplexity scores are not good even for large models (e.g., Llama2-7B, Llama2-13B), when compared to the recent quantization methods such as aespa [1], AutoRound [2], BoA [3], and GPTAQ [4]. Please compare the performance with these recent baselines, and also integrate BAQ into these methods to show the validity of the proposed BAQ.

2. A remaining concern is about the practicality. BAQ assigns different bits for each of weights, which is difficult to be implemented on NPU. For acceleration of such mixed-precision quantization, the authors need to develop their own CUDA kernel and show that quantization with different weight bits can also be accelerated as the standard uniform quantization. If the additional processing time is non-negligible, the the proposed method lacks practicality.

3. These days, integrating the quantization with outlier suppression techniques such as QuaRot and SpinQuant is the standard process for the quantization. Please show me the synergy with those techniques for Llama models.

[1] "Towards Next-Level Post-Training Quantization of Hyper-Scale Transformers", NeurIPS 2024.

[2] "Optimize Weight Rounding via Signed Gradient Descent for the Quantization of LLMs", EMNLP 2024.

[3] "BoA: Attention-aware Post-training Quantization without Backpropagation", ICML 2025.

[4] "GPTAQ: Efficient Finetuning-Free Quantization for Asymmetric Calibration", ICML 2025.

**Questions:**

See Weaknesses

---

### Official Review · Reviewer_r6vN · 2025-10-28

**Soundness:** 3
**Presentation:** 3
**Contribution:** 2
**Rating:** 4
**Confidence:** 5

**Summary:**

The paper introduces BAQ, a post-training quantization framework that optimally allocates bitwidths across weight columns in large language models (LLMs).
Based on the equal-loss principle, it formulates quantization as a convex optimization problem and derives a closed-form solution, enabling sensitivity-aware bit allocation that balances accuracy and compression under a fixed bit budget.

**Strengths:**

The method provides a theoretically grounded formulation of mixed-precision quantization through the equal-loss principle and derives an analytical optimality condition with a closed-form solution rather than relying on heuristics, making it a novel and well-established approach to bit allocation in LLM quantization.

**Weaknesses:**

- Although bits are allocated at the column level, it is unclear how this design leads to real hardware acceleration; the approach appears focused mainly on memory footprint reduction rather than compute efficiency. Hardware constraints and deployment aspects are not discussed, leaving practical feasibility uncertain.

-The comparative analysis is limited and lacks sufficient breadth. In particular, the paper omits comparisons with recent mixed-precision quantization methods that optimize under a fixed bit budget, making evaluations against only uniform-bit baselines not entirely fair. Although the granularity of mixed-precision application may differ, a more comprehensive qualitative and quantitative comparison with prior works that perform quantization under fixed bit or memory budgets is necessary for a fair and thorough evaluation.



https://arxiv.org/abs/2405.14917

https://arxiv.org/abs/2203.08368

https://arxiv.org/abs/2410.13056

https://arxiv.org/abs/2007.02017

Beyond this paper, numerous studies have explored mixed-precision quantization; therefore, a comprehensive survey, comparison, and categorization of these related works would be essential for a more complete analysis.

**Questions:**

- In Table 1, the proposed method underperforms SPQR on LLaMA-2 7B. Could you provide further analysis or insights into this result?


- Would it be feasible to extend the proposed bit allocation scheme from column-wise to layer-wise granularity? If so, what potential challenges or trade-offs (e.g., optimality, sensitivity estimation, or computational complexity) would arise in such an extension?

- (Optional) In BoA (https://arxiv.org/abs/2007.02017), performance gains are achieved through a more accurate Hessian approximation compared to GPTQ, whereas BAQ improves performance via column-wise mixed-precision bit allocation. Although these two approaches are orthogonal in nature, if one were to choose between them, which do you consider more efficient overall, taking into account factors such as computational cost, implementation complexity, and generalization across different model architectures?

**Details Of Ethics Concerns:**

.

---

### Official Review · Reviewer_6Smy · 2025-10-29

**Soundness:** 3
**Presentation:** 2
**Contribution:** 2
**Rating:** 2
**Confidence:** 4

**Summary:**

This paper proposes a weight-only mixed-precision quantization scheme (Bit Allocation Quantization, BAQ). It introduces a loss-bitwidth mapping that is leveraged to allocate bits, subject to a total budget, proportional to the sensitivity of each weight to the quantization scheme. Optimal allocation is justified theoretically. Results on LLMs up to 30B parameters show improved results in the regime of 2-bit compression against some competing algorithms for model compression (GTPQ, AWQ, SPQR).

**Strengths:**

- The paper is well written, well organized, and easy to follow
- The theoretical formulation to derive the optimal bitwidth allocation as a function of layer loss and sensitivity is elegant and well grounded
- The investigation encompasses large scale LLMs with 7-30B parameters, highly relevant to practical workloads nowadays
- An interesting link is drawn between dispersion of the sensitivity coefficients and BAQ effectiveness
- The authors share their code and analytical derivations

**Weaknesses:**

- A key limitation is that, contrary to what is stated, the technique is very _unfriendly_ to hardware as it requires assignment of independent bitwidths to each matrix column. This is not supported in today's accelerators. Running kernels where each column uses a different bitwidth would require custom per-column packing/unpacking logic, adding latency and potentially memory overhead
- In the 3-bit regime, perplexity/accuracy results are comparable to GPTQ. At 2 bits, although BAQ improves significantly on GPTQ, the resulting models still perform very poorly, showing large degradation against their unquantized counterpart, to the point of remaining, in practical terms, unusable
- There's a glaring lack of comparison (perplexity / accuracy) against state-of-the-art compression algorithms, such as OmniQ [1] and QuIP# [2], that were shown to significantly outperform GPTQ at high compression (2-3 bits)
- Table 1 does not show the 16-bit baselines
- The theoretical framework assumes continuous bitwidths $R_{ij}$, but for practical reasons the experiments round these bits allocations to the nearest bit, hence losing the analytically-derived guarantee of optimal allocation
- Fig. 3 may have been better presented as a scatter plot of Ratio_L vs Ratio_C across all weights in a given model (or multiple models), instead of showing only 4 layers

[1] https://arxiv.org/pdf/2308.13137
[2] https://arxiv.org/pdf/2402.04396

**Questions:**

- How do BAQ results compare against other compression algorithms?
- In what format could the weights quantized to different per-column bitwidths be stored?
- What are the expected overheads associated with decompression at inference time?

---

### Official Review · Reviewer_cJWm · 2025-11-01

**Soundness:** 2
**Presentation:** 2
**Contribution:** 2
**Rating:** 2
**Confidence:** 4

**Summary:**

The paper presents a precision-allocation rule for post-training weight quantization.

**Strengths:**

+ The new formulation could be of interest.

**Weaknesses:**

- Fine-grain, ungrouped mixed precision quantization is not hardware-acceleration friendly, limiting the practical value, which is not convincingly established empirically here either.

**Questions:**

* How can this be generalized to scenarios with activation-quantization?

---

### Note · Authors · 2025-11-25

I have read and agree with the venue's withdrawal policy on behalf of myself and my co-authors.